# The Performance of Using the Parasympathetic Tone Activity (PTA) Index to Assess Intraoperative Nociception in Cats

**DOI:** 10.3390/vetsci11030121

**Published:** 2024-03-06

**Authors:** Leonor Lima, José Diogo Dos-Santos, Lénio Ribeiro, Patrícia Cabral, Bruno Colaço, João Martins

**Affiliations:** 1Faculty of Veterinary Medicine, Lusófona University, 1749-024 Lisbon, Portugal; 2Veterinary and Animal Science Research Centre (CECAV), Faculty of Veterinary Medicine, Lusófona University—Lisbon University Center, 1749-024 Lisbon, Portugal; 3Veterinary and Animal Science Research Centre (CECAV), UTAD, 5000-801 Vila Real, Portugal; 4Associate Laboratory for Animal and Veterinary Sciences (AL4AnimalS), 1300-477 Lisbon, Portugal; 5Research Center for Biosciences and Health Technologies (CBIOS), Universidade Lusófona de Humanidades e Tecnologias, 1749-024 Lisbon, Portugal; 6Centre for the Research and Technology of Agro-Environmental and Biological Sciences (CITAB), UTAD, 5000-801 Vila Real, Portugal

**Keywords:** animal welfare, cat, heart rate, nociception, parasympathetic tone activity

## Abstract

**Simple Summary:**

The monitoring of nociception/antinociception poses a significant challenge during anesthesia in cats. Parasympathetic Tone Activity (PTA) monitoring could prove to be a useful tool for this purpose; however, there is no research on this species. This study aims to assess the effectiveness and speed of PTA monitoring in detecting nociception when compared to heart rate (HR) during the intraoperative period in female cats undergoing ovariectomy (OV). The monitoring included intraoperative instantaneous PTA (PTAi), heart rate (HR), respiratory rate (*f*_R_), non-invasive systolic and mean arterial pressure (SAP/MAP) (mmHg), and the time required for HR (HR time) and PTAi (PTAi time) to reach their minimum peak values after the application of five surgical stimuli (SS). Each surgical stimulus was categorized into three groups: no nociception detection with no HR reaction and a PTAi > 50 (*Nhre*), no HR reaction and a PTAi < 50 (*Nhre* < 50), and a HR reaction and a PTAi < 50 (*Hre* < 50). The findings of this study suggest that in a clinical setting, PTA monitoring may be more effective in detecting nociception compared to HR, and with more intense SS, both parameters present a similar speed in detecting nociception.

**Abstract:**

Background: The monitoring of nociception/antinociception poses a significant challenge during anesthesia, making the incorporation of new tools like the Parasympathetic Tone Activity (PTA) monitor an added value in feline anesthesia. Objectives: To compare the effectiveness and speed of PTA monitoring when compared to heart rate (HR) in detecting surgical stimuli (SS) during the intraoperative period in 49 female cats undergoing ovariectomy (OV). Methods: Instantaneous Parasympathetic Tone Activity (PTAi) values, HR, *f*_R_, and non-invasive SAP and MAP were continuously monitored and systematically assessed. The time required for HR (HR time) and PTAi (PTAi time) to reach their minimum peak values following each surgical stimulus was collected at five points for each anaesthetized cat. Each collected surgical stimulus was categorized into 3 groups for statistical analysis: no nociception detection, no hemodynamic reaction and a PTAi > 50 (*Nhre*); no hemodynamic reaction and a PTAi < 50 (*Nhre* < 50); and hemodynamic reaction and PTAi < 50 (*Hre* < 50). Results: PTAi response demonstrated effectiveness in detecting nociception compared to HR. The SS were categorized as 36.1% in the *Nhre* group, 50% in the *Nhre* < 50 group, and only 13.9% in the *Hre* < 50 group. In the *Hre* < 50 group, PTAi time and HR time had similar speeds in detecting the SS (*p* = 0.821); however, PTA time was significantly slower in the *Nhre* < 50 group when compared to the *Hre* < 50 group (*p* = 0.001). Conclusions: PTA monitoring may be a useful tool to complement HR monitoring for detecting nociception. PTA monitoring demonstrated a superior diagnostic value compared to HR for detecting nociception in cats undergoing OV and a similar speed to HR in detecting SS when HR increases above 20%. Future studies are needed to understand in a clinical setting the meaning of sympathetic activation/nociception detected using the PTA monitor when the HR increase is not clinically relevant.

## 1. Introduction

The monitoring of nociception/antinociception poses a significant challenge during anesthesia, as there is no objective and absolute measure to quantify nociception [1,2]. The most common response to “surgical stress” is an increase in sympathetic activity or a corresponding decrease in parasympathetic tone [3].

When assessing nociception in anesthetized animals, our focus lies on detecting hemodynamic reactivity, which is related to increased blood pressure and tachycardia, as well as alterations in respiratory patterns or muscle tone [4]. However, these modifications are not necessarily and explicitly related to nociception [5], as they can be influenced by a light depth of anesthesia, vagal stimulation [6], and the anesthetic agents administrated [4,6]. It is widely recognized that inadequate analgesia leading to nociception can result in complications during the perioperative period [7], discomfort during recovery, and persistent surgically induced neuropathic pain, as experienced by 10–50% of humans following common surgical procedures [8,9]. Therefore, it is essential to incorporate new tools capable of assessing nociception during the intraoperative period. Among these tools, nociception monitors in human anesthesia [10,11] emerge as a prominent example, demonstrating substantial advantages and the potential to be integrated into closed-loop systems for the administration of analgesics [7,12,13], enabling anesthetists to practice using and pre-emptively adjust anti-nociceptive medication levels, thereby preventing stress-associated hemodynamic changes [14].

The Analgesia Nociception Index (ANI) is a new monitor utilized in human medicine to quantify the balance between nociception and antinociception [5,11,15,16], while in veterinary medicine, Parasympathetic Tone Activity (PTA) serves a similar purpose [17,18,19]. The PTA signal is acquired in real time using a lead II electrocardiogram (ECG) [18]. PTA is validated for intraoperative nociception detection and is based on analysis of the heart rate variability by capturing the rapid increments in parasympathetic tone resulting from each respiratory cycle (spontaneous or artificial) to measure the “relative quantity” of parasympathetic tone [18,20]. Rapid changes in the parasympathetic tone are expressed in the sinus node of the heart as alterations in the time intervals between two R waves of an electrocardiogram [3,18]. Strong nociceptive stimulation elicits an almost immediate reduction in the amplitudes of the respiratory modulations observed in the RR series, attributed to the rapid decrease in parasympathetic tone. PTA values range from 0 and 100. In anaesthetized patients, optimal values are defined as scores between 50 and 70, suggesting the absence of nociception [17,19,21]. PTA values close to 100 correspond to a predominant parasympathetic tone or opioid overdose, while values below 50 correspond to a predominant sympathetic tone associated with a high level of stress or nociception [12,19]. Values <50 suggest an absence of analgesic balance, and therefore the presence of nociception should be considered [17,19]. The monitor continuously displays two PTA values that are updated every second: the instantaneous PTA index (PTAi), derived from the calculation based on the preceding 54 PTA values, and the averaged PTA (PTAm), derived from the calculation based on the preceding 176 PTA values [22].

Despite PTA demonstrating a significant finding in nociception monitoring, its widespread utilization remains limited due to a scarcity of data [23], particularly in the context of feline patients.

This study aims to evaluate the usefulness of the PTA monitor in detecting nociception after surgical stimuli (SS), when compared to heart rate (HR), in female cats undergoing ovariectomy (OV). The study hypothesized that the PTA monitor would be more effective and faster in detecting SS when compared with HR.

## 2. Materials and Methods

### 2.1. Animals

A total of 49 female cats weighing 3.05 ± 0.64 kg [mean ± standard deviation (SD)] spanning various age groups under institutional government service care were submitted to a stray animal sterilization program and underwent elective OV using a median laparotomy. The protocol of this study was approved by the Institutional Animal Care and Use Committee of Lusófona University (no. 15/2022). This was a clinical, post hoc, randomized, blinded study conducted between November and December of 2022. The animals were captured the day before surgery. Only animals with ASA I or II classification scores (American Society of Anesthesiologists) were included in the study. This score was obtained by evaluating the physical examination findings, complete blood cell count, and serum biochemical analysis results.

The exclusion criteria consisted of animals receiving pharmacological agents affecting the sympathetic or parasympathetic nervous system throughout the study (atropine, glycopyrrolate, ephedrine, epinephrine, or norepinephrine), animals exhibiting an absence of sinus cardiac rhythm, and failures in both the PTA monitor and signal capture.

### 2.2. Anaesthesia and Monitoring

All cats received an intramuscular (IM) injection of dexmedetomidine (20 μg kg^−1^; Dexdomitor 0.5 mg mL^–1^, Orion, Finland) and methadone (0.2 mg kg^−1^; Semfortan 10 mg mL^–1^, Dechra, Italy) as premedication. Blood samples were collected from the jugular vein, and a catheter was introduced into the cephalic vein under aseptic conditions for drug and fluid administration. The cats were pre-oxygenated with 5 L min^–1^ of oxygen 100% using a facemask.

Anesthesia was induced with propofol (Propofol 10 mg mL^–1^; B. Braun Medical Inc., Germany) given intravenously (IV), at a dose of 2–4 mg kg^−1^, titrated to effect. Orotracheal intubation was performed using an appropriately sized cuffed orotracheal tube after the topical application of 0.1 mL of 2% lidocaine to the larynx. The cats were then transferred to the surgical theatre and positioned in dorsal recumbency. The cats were allowed to breathe spontaneously while being connected to a non-rebreathing system (Ayre’s T-Piece breathing circuit) connected to an anesthetic machine (WATO EX-20, Mindray, Shenzhen, China). anesthesia was maintained with isoflurane for the duration of surgery (IsoFlo; Zoetis, Madrid, Spain) at 1 ± 0.1% using 250 mL kg^−1^ per minute of oxygen as the carrier gas.

During the perioperative period, Lactated Ringer’s solution (Lactated RingerVet; B. Braun Medical Inc., Germany) was administered intravenously, started at a rate of 3 mL kg^−1^ per hour, using an infusion pump (Infusomat; B. Braun Medical Inc., Melsungen, Germany). Heart rate (HR; beats min^−1^) and rhythm, respiratory rate (*f*_R_; breaths min^–1^), end-tidal carbon dioxide (EtCO2; mmHg), inspiratory fraction of carbon dioxide (FiCO2; mmHg), esophageal temperature (°C), and capillary oxygen hemoglobin saturation (SpO2; %) were continuously monitored. For non-invasive blood pressure measurement, an oscillometric method with an appropriately sized cuff placed around the right or left antebrachium was used to measure systolic and mean arterial pressure (SAP/MAP) (mmHg) (multiparametric monitor BeneVision N15; Mindray, Shenzhen, China). Body temperature was monitored and maintained within an optimal range (36.5 °C to 38.5 °C) using a heating blanket (Thermal Blanket Carbonvet cage, B. Braun Medical Inc., Shanghai, China). Cats that showed an HR increase exceeding 20% following SS (hemodynamic reaction) immediately received rescue analgesia in the form of an IV bolus of fentanyl (Fentadon; 50 ug mL^–1^, Dechra, Italy) at a dose of 4 μg kg^−1^. At the end of each procedure, isoflurane was discontinued, and following extubation, each cat received 0.1 mg kg^−1^ of meloxicam subcutaneously (SC) (Meloxidyl, Ceva Santé Animale, Libourne, France), as well as 25 ug kg^−1^ of an atipamezole IM (Antisedan, Orion Corporation, Espoo, Finland) for the reversal of the sedative effects of dexmedetomidine. All the OVs were performed by two experienced surgeons. The surgical technique is described elsewhere.

### 2.3. PTA Monitoring

The PTA index measurement was performed using a lead II ECG signal after placing three electrode gel-moistened flattened crocodile clips attached to the skin at the level of the olecranon on the caudal aspect on each of the thoracic limbs and over the patellar ligament on the cranial aspect of the right pelvic limb [23].

The criteria for considering a PTAi measurement valid included the good signal quality of the monitor recording and the presence of a minimum value comprising between 50 and 70 [4] before each surgical stimulus.

### 2.4. Study Design

The study is based on the evaluation of nociceptive stimuli during surgery. Five pre-determined SS that typically activate nociception were evaluated per cat (S1, S2, S3, S4, S5). S1 represents the clamping of surgical field drapes to the animal using Backhaus clamps. S2 involves a cutaneous incision to the abdominal cavity. S3 represents traction and handling of the right ovary. S4 corresponds to traction and handling of the left ovary. Lastly, S5 is associated with cutaneous suturing of the abdominal cavity.

During a medium-depth plane of anesthesia, the baseline values for the HR, *f*_R_, SAP, MAP, and PTAi were established. Subsequently, five SS were conducted in all cats. The time elapsed since the initiation of each surgical stimulus until reaching the minimum peak, immediately before the start of the PTAi ascent, was collected. The time and value data were acquired using the PTA monitor software. In cases where the heart rate exceeded 20% of pre-stimulation (HR time), the duration was measured using a digital chronometer and recorded.

The criterion for a positive nociceptive response was defined as a PTAi value < 50, either with or without a change equal to or exceeding 20% of pre-stimulation of the HR. This value was adapted from human medicine [12] and pigs [19], where a PTAi value below this cut-off was considered nociception [12]. The SS were categorized into 3 groups: no nociception detection with no hemodynamic reaction and a PTAi > 50 (*Nhre*), no hemodynamic reaction and a PTAi < 50 (*Nhre* < 50), and a hemodynamic reaction and a PTAi < 50 (*Hre* < 50) (Figure 1).

### 2.5. Statistics

The statistical analysis was performed using SPSS 29.0 for Windows (IBM SPSS Statistics, New York, NY, USA). The Shapiro–Wilk test was used to test the data normality. The data were expressed as mean ± standard deviation (SD). Analysis of variance (ANOVA) using the Bonferroni post hoc test was applied to detect the differences in the PTAi, HR, *f*_R_, SAP, and MAP means between different groups. An independent t-test was applied to compare the differences in the PTAi time observed between the hemodynamic reaction (*Hre* < 50) and no hemodynamic reaction (Nhre < 50) groups. A paired-samples t-test was applied to compare the differences between PTAi time and HR time. Based on a power of 0.8, an effect size of 0.25, and an alpha error of 0.05, 158 valid PTAi measurements were sufficient to detect a relevant percentage change in the PTAi (G*Power 3.1.9.7.). Therefore, considering a possible technical failure of 30%, a sample size of 49 cats (245 PTAi measurements) was selected. A *p*-value < 0.05 indicates statistical significance.

## 3. Results

Out of the 49 cats included, all variables were accurately measured for 166/245 (67.8%) SS. The remaining 79/245 (32.2%) SS were excluded due to PTAi technical failures (Figure 1). Among these 166 nociceptive stimuli, 60/166 (36.1%) were included in the *Nhre* group, 83/166 (50%) in the *Nhre* < 50 group, and 23/166 (13.9%) in the *Hre* < 50 group (Figure 1).

A decrease in the PTAi below 50 was observed in 106/166 (63.9%) SS, with *n* = 4 for S1, *n* = 8 for S2, *n* = 36 for S3, *n* = 32 for S4, and *n* = 25 for S5. The mean PTAi values were significantly different in the *Nhre* group (*p =* 0.001) compared to both the *Nhre* < 50 and *Hre* < 50 groups (Figure 2).

The time needed for the SS to induce a decrease in the PTAi below 50 was significantly different between the *Nhre* < 50 group (79.2 ± 84.5 s) and the *Hre* < 50 group (35.3 ± 32.4 s) (*p* = 0.001). It is also possible to verify that the *Nhre* < 50 group showed more dispersed values and a longer PTAi time in comparison to the *Nhre* < 50 group (Figure 3).

No statistically significant difference was observed between the PTAi time (34.9 ± 24.7 s) and HR time (34.2 ± 14.8 s) in the *Hre* < 50 group (*p* = 0.693) (Figure 4).

Related to *f*_R_, statistically significant differences were observed between all groups (Table 1). A statistically significant difference was observed in the SAP between the *Hre* < 50 group and the other groups. No statistically significant difference was observed in the MAP between the three groups (Table 1). The values included in Table 1 were obtained during general anesthesia from baseline to the last point analyzed. These results were assessed after the post hoc analysis and formation of the groups.

## 4. Discussion

The main findings of the current study carried out in a clinical setting in cats undergoing elective OV suggest that the PTA monitor, when using a cut-off value of PTAi < 50 for nociception, demonstrated superior effectiveness in detecting sympathetic nervous system activation/nociception when compared to HR, which is in line with the results of studies carried out in dogs and pigs [12,19,24] and humans [5]. PTA monitoring detected 106 (63.9%) of SS that corresponded to sympathetic nervous system activation, and the HR only detected 23 (13.9%) SS. The question arises as to whether the application of the existing recommendations advocating for approximately 20% increments in the HR during intraoperative anesthesia is sufficiently effective in achieving adequate control of intraoperative nociception. In our study, HR was not effective in detecting 83 (50%) SS, meaning that they were not considered clinically relevant, and no rescue analgesia was applied.

The effectiveness of the PTA monitoring in detecting nociception in cats in 83 SS for the *Nhre* < 50 group is supported by the fact that, despite the absence of a 20% or higher increase in HR, they were significantly higher when compared to the mean values of the *Nhre* group.

PTAi values less than 50 were predominantly observed in two specific SS corresponding to the traction and handling of both ovaries, specifically S3 (right ovary) and S4 (left ovary), which are procedures widely acknowledged to induce nociceptive stimulation [25]. Our study shares similarities with an investigation conducted in dogs, yet it has the advantage that the examined SS had both visceral and somatic surgical components, thus increasing the clinical significance of this study [17].

Statistically significant differences were observed in our study regarding PTAi time and HR time. In the *Nhre* < 50 group, the PTAi time (79.2 ± 84.5 s) was significantly slower than the PTAi time (35.3± 32.4 s) in the *Hre* < 50 group. These results may indicate that the SS included in the *Nhre* < 50 have a lower intensity than those included in the *Hre* < 50 group. The PTA time was faster, though not significantly different from the HR time (37.5 ± 18.8 s) in the *Hre* < 50 group. These findings align with a study in dogs by Mansour et al., 2017 [17] but are different from those reported Aguado et al., 2020 [6]. In the Aguado study, no differences in the PTAi response time were observed in the medium- and high-stimuli groups [6]. These discrepancies likely arise from methodological distinctions, particularly in the drugs administered before applying nociceptive/surgical stimuli and the nature of the nociceptive/surgical stimuli themselves. Aguado et al., 2020, exclusively investigated somatic nociception, while our study evaluated both somatic and visceral nociception [6]. This difference resulted in the assessment of distinct nerve pathways.

Our findings are consistent with other clinical studies conducted in humans where the ANI was used to assess the nociception response following a nociceptive stimulus, with its dynamic variation allowing for the anticipation of hemodynamic responses associated with nociception. According to the ANI in human patients, it has been observed that modifications in the heart rate variability (HRV) precede hemodynamic responses to nociceptive stimulation [26,27]. A study in dogs showed that a PTA below 48 increased the probability of a hemodynamic response, and a fall of 18% predicted with an accuracy of 80% a hemodynamic response [17].

In our study, *f*_R_ presented a significant increase in the *Nhre* < 50 and *Hre* < 50 groups. These results may suggest potential sympathetic nervous system activation/nociception whenever the PTAi drops below 50.

The SAP presented a significant increase in the *Hre* < 50 groups. Regarding MAP, no significant differences were observed between groups. The lack of an invasive pressure line precluded real-time assessment of the hemodynamic impact during SS. Nevertheless, it is expected that a hypertensive episode will persist long enough to be detected using the oscillometric method used in this study. To confirm our results, further investigation evaluating invasive blood pressure in cats should be conducted.

The autonomic nervous system’s response to nociception has several consequences for the cardiovascular, gastrointestinal, immunological, endocrine, and metabolic levels in patients [4], influencing both the stability of anesthesia and the quality of recovery [28]. Nociception control decreases chronic postsurgical pain [29] and reduces postoperative morbidity and mortality [30]. Determining whether the activation of the sympathetic system, as detected using nociception monitors, particularly the PTA monitor, exclusively identifies nociception or whether it is influenced by other mechanisms and pathways activating the sympathetic nervous system unrelated to nociception, such as the anesthetic plan or the influence of drugs used, is key [31]. In the future, assessing the feasibility of applying PTA monitoring in feline anesthesia to managing the requirement for rescue analgesia will be crucial, although this was not the focus of this study. In our study, sympathetic nervous system activation (PTAi < 50) was not always accompanied by a clinically significant increase in HR (>20%), i.e., the need for rescue analgesia. This aligns with the findings in human reports [5]. According to a study conducted on children, the primary benefit of a nociceptive monitor is to optimize opioid titration to prevent overdosage and underdosing, which can lead to opioid-induced hyperalgesia or postoperative pain, respectively [6].

The current study has several limitations that must be considered. One notable limitation is the involvement of two distinct surgeons, which introduces the possibility of individual variations. In addition, the presence of two data collection operators can also be considered a potential limitation. It is worth noting that non-invasive blood pressure monitoring may not capture immediate changes in arterial pressure, and therefore future studies including invasive blood pressure monitoring to confirm these results are warranted. In future studies, it may be interesting compare PTAi values with the variation in arterial pressure and fR, in addition to HR [32]. Being a study carried out in a clinical setting, the administration of rescue analgesia with fentanyl may have introduced bias into the PTAi and HR parameter monitoring. One of the limitations of this investigation is the absence of the independence of the surgical stimuli collected from each cat, potentially leading to the initial stimulus interfering with subsequent ones. Despite wanting the PTAi value to normalize (PTAi > 50), before each stimulus, similar to Aguado et al. 2020’s methodology in dogs, the recovery time between SS was not standardized [6]. Grouping the data allowed us to understand the strict relationship between PTAi and HR activation. Particularly in the *Nhr* < 50 group, it was revealed that 83 SS instances correlated with a PTAi decrease below 50, without a corresponding HR increase exceeding 20%. Despite the statistically significant increase in HR, conducting an analysis based on surgical stimulation or combined data would not enable the assessment of the relationship between PTAi and HR. Pre-medication with dexmedetomidine and methadone could have influenced the hemodynamic response, potentially leading to a reduction in HR. In future studies, it is important to explore the interaction with other drugs. Additionally, the involvement of two surgeons in the surgical procedure adds an element of unpredictability to the nociceptive response and gives potential variation in the manipulation techniques. Another important limitation is related to the nature of our study population, as the cats included in the study were sourced from colonies or were stray cats. It would also be important to establish a correlation between pain scale scores and our intraoperative nociceptive stimuli groups to understand whether animals that manifested a high degree of nociception during the surgical procedure were associated with elevated pain scale scores. One of the main problems in this study was the difficulty of obtaining a reliable signal using the PTA monitor, as 79 excluded PTA values came from different animals and were due to signal capture failures, thereby impeding the start of surgery and data collection for the animals in question. Consequently, there were instances where the manufacturer’s predefined placement of ECG clips was not strictly followed (yellow sensor on left forelimb; red sensor on right forelimb; black sensor on right hindlimb). It would be important in future studies to confirm whether the difficulty of acquiring an adequate signal is related to the feline species.

As one of the few clinical studies conducted in cats, the present study aimed to address the notable lack of data concerning PTA in this species, in which we observed clinically significant outcomes with its use. It would be interesting to evaluate nociception using the same PTA monitor in different surgical contexts and types and/or with different drugs administrated in cats [33] and deepen the study of the relationship between PTAi and the remaining hemodynamic parameters.

## 5. Conclusions

The PTA monitor may be a useful tool for detecting the activation of the sympathetic nervous system related to nociception in feline anesthesia. PTA monitoring demonstrated a superior diagnostic value to HR for detecting sympathetic nervous system response/nociception in cats undergoing OV under the used protocol with dexmedetomidine and methadone. Also, PTA monitoring detects surgical stimuli more quickly when the HR increases above 20%, suggesting that higher-intensity stimuli are associated with faster activation of the sympathetic nervous system. Future studies are needed to understand in a clinical setting the meaning of sympathetic activation/nociception detected using the PTA monitor when the HR increase is not clinically relevant.

## Figures and Tables

**Figure 1 vetsci-11-00121-f001:**
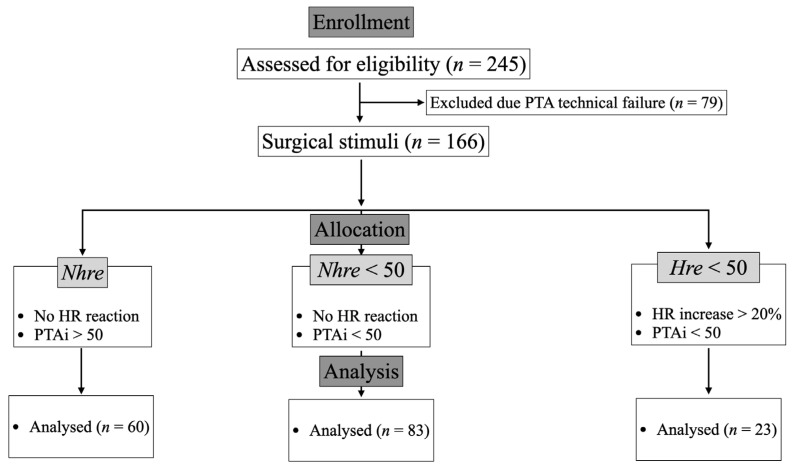
Consort design diagram of the study. HR = heart rate; Hre = hemodynamic reaction; Nhre = no hemodynamic reaction; PTAi = Instantaneous Parasympathetic Tone Activity.

**Figure 2 vetsci-11-00121-f002:**
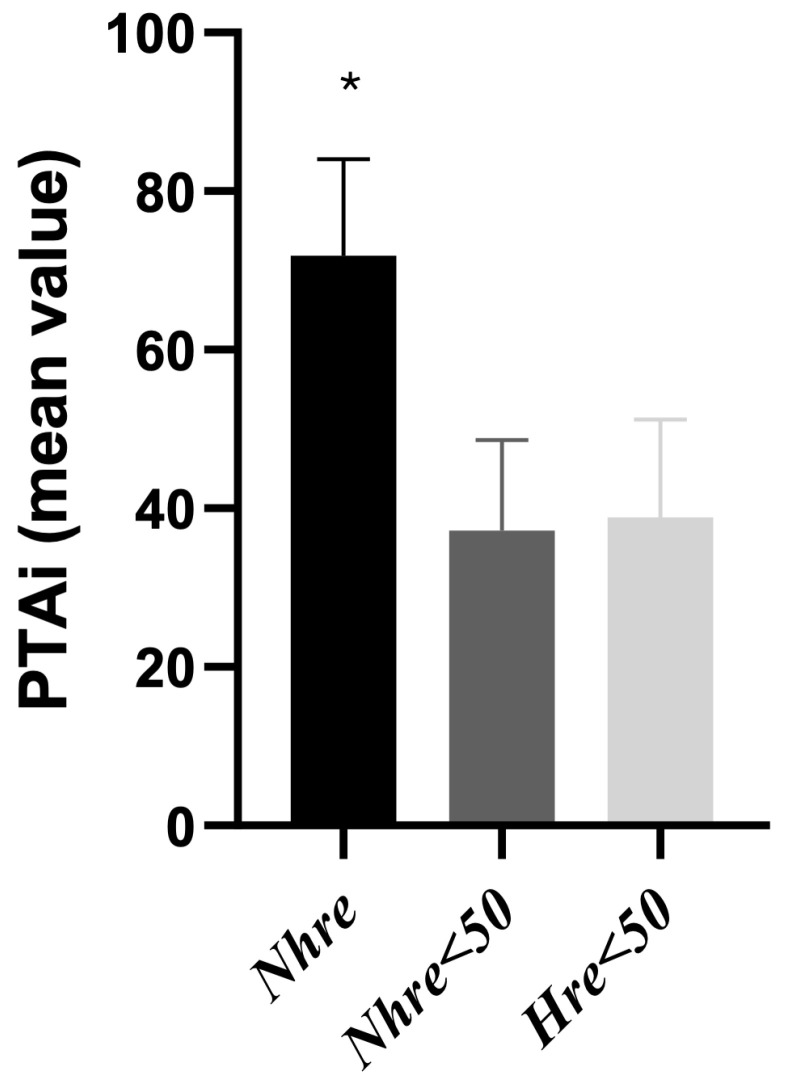
The Instantaneous Parasympathetic Tone Activity (PTAi) mean values were obtained in different groups. The mean PTAi values were significantly different in the *Nhre* group (*p =* 0.001) compared to both the *Nhre* < 50 and *Hre* < 50 groups. Hre = hemodynamic reaction; Nhre = no hemodynamic reaction; PTAi = Instantaneous Parasympathetic Tone Activity. * indicates statistical significance (*p*-values < 0.05).

**Figure 3 vetsci-11-00121-f003:**
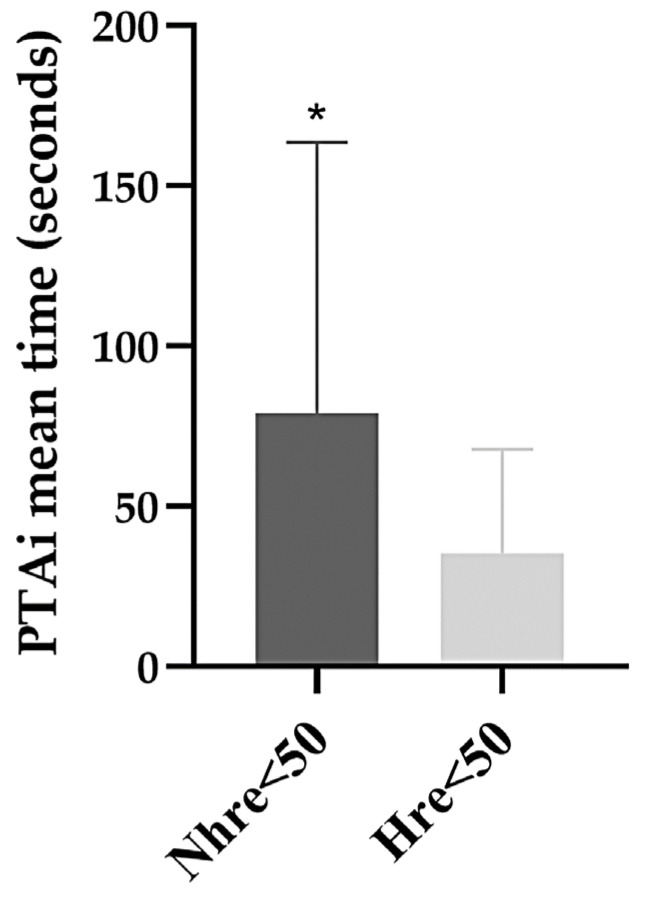
The time of Instantaneous Parasympathetic Tone Activity (PTAi) observed between PTAi time in hemodynamic reaction (*Hre* < 50) and no hemodynamic reaction (Nhre < 50) groups after surgical stimuli application. Hre = hemodynamic reaction; Nhre = no hemodynamic reaction; PTAi = Instantaneous Parasympathetic Tone Activity. * indicates statistical significance (*p*-values < 0.05).

**Figure 4 vetsci-11-00121-f004:**
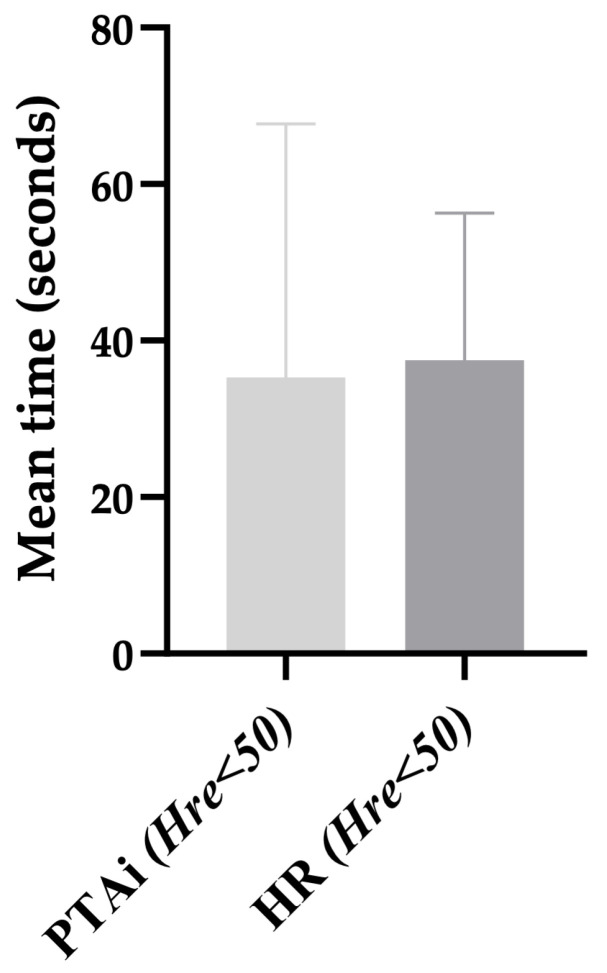
The time of Instantaneous Parasympathetic Tone Activity and heart rate time in the hemodynamic reaction and PTA < 50 (*Hre* < 50) group after surgical stimuli application. HR = heart rate; *Hre* = hemodynamic reaction; PTAi = Instantaneous Parasympathetic Tone Activity.

**Table 1 vetsci-11-00121-t001:** Means and standard deviation for *f*_R_, SAP, and MAP in the three groups (*Nhre, Nhre* < 50, *Hre* < 50). Mean values with different superscripts (a, b, c) are statistically different in the ANOVA post hoc Bonferroni test (*p* < 0.05). *f*_R_ = respiratory rate; Hre = hemodynamic reaction; MAP = median arterial pressure; Nhre = no hemodynamic reaction; SAP = systolic arterial pressure.

	*Nhre*	*Nhre* < 50	*Hre* < 50	*p*-Value
*f* _R_	20.6 ± 7.43 ^a^	24.7 ± 8.22 ^b^	29.8 ± 7.1 ^c^	0.001
SAP	109.23 ± 20.7 ^a^	106.4 ± 15.6 ^a^	121.9 ± 19.64 ^b^	0.002
MAP	81.5 ± 19.4 ^a^	79.2 ± 14.8 ^a^	86.43 ± 20.4 ^a^	0.204

## Data Availability

The data are contained within the article.

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
