# Peer review of "The Performance of Using the Parasympathetic Tone Activity (PTA) Index to Assess Intraoperative Nociception in Cats"

_vetsci, 2024, doi:10.3390/vetsci11030121_

Round 1

Reviewer 1 Report

Comments and Suggestions for Authors

Dear authors,

this is a very interesting study, well explained, with sections presented in a balanced and coherent way. Introduction is properly documented, and the data are properly discussed. I suggest some revision providing some comments:

SIMPLE SUMMARY

You have to improve this part. For example, you should better specify how your study could be valuable for the veterinary world and for the society.

ABSTRACT

You must be discursive and less schematic (eliminating objectives, materials and methods, etc...). Add the number of dogs used in this study. Explain better the meaning of PTA.

INTRODUCTION

Line 47 : “in our animals?” Please rewrite.

MATERIALS AND METHODS

Why were 49 animals selected? Has been done an evaluation of the sample size? How many animals were excluded from the study before surgery and why?

During the classical technique did you use analgesic techniques to minimize intraoperative pain (incisional anesthesia or splash block)?

Please add these recent references:

Cicirelli V, Burgio M, Lacalandra GM, Aiudi GG. Local and Regional Anaesthetic Techniques in Canine Ovariectomy: A Review of the Literature and Technique Description. Animals (Basel). 2022 Jul 27;12(15):1920. doi: 10.3390/ani12151920. PMID: 35953908; PMCID: PMC9367435.

Author Response

Many thanks for your e-mail concerning our article. We have improved our work to attend to all comments. All changes in the manuscript are highlighted in yellow.

Reviewer 2 Report

Comments and Suggestions for Authors

This is, in general, a well written report on a well planned study that provides important new information relating to nociception during anaesthesia in cats. I have only one major comment and suggestion.

Obviously PAC is correlated with HR. Incidentally why do you start by defining HR as heart rate, then switch to the ponderous expression haemodynamic reaction, which adds nothing and serves only to confuse. I recognise that the PAC response appears to be more sensitive and more raped in onset. I suggest it might be interesting to present scattergrams of HR v. PAC at two intervals after S3 and S4.  These could illustrate the magnitude of the advantage in the use of PAC. Alternatively it could indicate that HR monitoring might be sufficient for vets without access to high-tech monitors.

Comments on the Quality of English Language

good

Author Response

(The authors gave the same response as above.)

Reviewer 3 Report

Comments and Suggestions for Authors

Thank-you for this interesting paper, and I would like to commend you on the clarity of the writing - on that aspect, I had only a few minor test suggestions which I will detail below.

However, I do have some issues with the statistical approach. As each SS was performed on each cat in series, the data points are not independent, and as such you need to use a pseudoreplication/repeated-measures structure in your analysis. It is likely that the impact of each SS is to some extent cumulative (which is suggested by Figure 2). Within the model you also need to fit 'surgeon' and 'observer' as fixed effects.

The potential for a cumulative effect will also influence your results pertaining to 'time to peak response', so you need to nest the data within SS, and consider the non-independence of observations.

Your key finding is that PTAi is better able to detect nociception - that would be based on the proportion of nociceptive stimuli eliciting a response - however, the data underpinning this is buried in an almost throwaway comment at the start of the results section. (also not made very much of in the discussion - sad for a key conclusion, and it makes the rest of the discussion seem a little irrelevant, as it is not supporting the key conclusion) The proportions haven't even been compared e.g. using Chi-square test, and again there is no consideration of non-independent observations.

The analysis of the data presented in table 1 is fundamentally flawed. You assigned animals to groups depending on the data generated... of course there are differences!

A better approach would be non-linear modelling, looking at probability of detecting each type of SS with each of the measures used (and/or the combined 'any Hre' metric). Correlations between the different Hre would also be of interest.

My recommendation is to reanalyse these data - I think you may find some more interesting and useful aspects to report.

Text suggestions:

Line 18 - "(PTA) measures the"

Line 19 - "tone and values"; "this study aims to"

Line 20 - "monitoring and conventrional"

Line 21 - delete "For that a"

Line 22 - "five SS of each anaestetized cat"

Line 28 - "PTA monitoring and"

Line 29 - "in 49 female cats"

Line 34 - "(SS) of each anaesthetized cat."

Line 95 - what does "under * care" mean?

Line 170 - clarify that these parameters are abbreviated to "HR time" and "PTAi time"

Line 177, should that second group read ", No hemodynamic reaction and PTAi <10 (Nhre<50)

Line 322 - "dogs, pigs and humans"

Author Response

(The authors gave the same response as above.)

Round 2

Reviewer 3 Report

Comments and Suggestions for Authors

Dear authors, thank-you for your revisions, the paper is much clearer now.

Author Response

(The authors gave the same response as above.)
